# Proteome and Glycoproteome Analyses Reveal the Protein N-Linked Glycosylation Specificity of STT3A and STT3B

**DOI:** 10.3390/cells11182775

**Published:** 2022-09-06

**Authors:** Piaopiao Wen, Jingru Chen, Chenyang Zuo, Xiaodong Gao, Morihisa Fujita, Ganglong Yang

**Affiliations:** 1Key Laboratory of Carbohydrate Chemistry and Biotechnology, School of Biotechnology, Jiangnan University, Ministry of Education, Wuxi 214122, China; 2Institute for Glyco-Core Research (iGCORE), Gifu University, Gifu 501-1193, Japan

**Keywords:** STT3A, STT3B, N-linked glycosylation, glycoproteomics, intact glycopeptides, mass spectrometry

## Abstract

STT3A and STT3B are the main catalytic subunits of the oligosaccharyltransferase complex (OST-A and OST-B in mammalian cells), which primarily mediate cotranslational and post-translocational N-linked glycosylation, respectively. To determine the specificity of STT3A and STT3B, we performed proteomic and glycoproteomic analyses in the gene knock-out (KO) and wild-type HEK293 cells. In total, 3961 proteins, 4265 unique N-linked intact glycopeptides and 629 glycosites representing 349 glycoproteins were identified from all these cells. Deletion of the STT3A gene had a greater impact on the protein expression than deletion of STT3B, especially on glycoproteins. In addition, total mannosylated N-glycans were reduced and fucosylated N-glycans were increased in STT3A-KO cells, which were caused by the differential expression of glycan-related enzymes. Interestingly, hyperglycosylated proteins were identified in KO cells, and the hyperglycosylation of ENPL was caused by the endoplasmic reticulum (ER) stress due to the STT3A deletion. Furthermore, the increased expression of the ATF6 and PERK indicated that the unfolded protein response also happened in STT3A-KO cells. Overall, the specificity of STT3A and STT3B revealed that defects in the OST subunit not only broadly affect N-linked glycosylation of the protein but also affect protein expression.

## 1. Introduction

Protein N-linked glycosylation is one of the primary post-translational modifications that occur in the endoplasmic reticulum (ER) and has important effects on protein folding, degradation, stability, trafficking and cellular communication [1,2]. Lipid-linked oligosaccharides (LLO) consisting of dolichol-pyrophosphate-linked Glc3Man9GlcNAc2 (where Glc, Man, and GlcNAc are glucose, mannose, and N-acetylglucosamine, respectively) are biosynthesized by the stepwise addition of sugars to the dolichol-phosphate. The preassembled LLO is en bloc transferred to the acceptor sites (N-X-S/T, X is any amino acid except proline) on nascent polypeptides, which is mediated by oligosaccharyltransferase (OST) complexes. OST is a multi-transmembrane complex located on the ER membrane. In mammals, two major types of OST complexes exist: OST-A and OST-B. Six OST subunits (ribophorin-I, ribophorin-II, OST48, TMEM258, DAD1, and OST4) are present in both OST-A and OST-B [3,4]. In addition, there are three unique subunits (STT3A, DC2, KCP2) found only in OST-A that interact with the protein translocation channel Sec61 to regulate cotranslational glycosylation [5,6]. On the other hand, OST-B contains STT3B and MAGT1 or TUSC3 as specific subunits [7]. OST-B is primarily responsible for post-translocational glycosylation, especially the sites skipped by OST-A. STT3A and STT3B are the catalytic subunits of the OST-A and OST-B complexes, respectively. Recently, several studies have clarified that STT3A and STT3B proteins have different preferences for potential glycosites. For example, Shrimal et al. conducted pulse labeling experiments on five human glycoproteins and found that residues 50–55 at the extreme C-terminal regions are modified by STT3B-dependent post-translocational glycosylation, and glycosylation of the C-terminal NXT site occurs faster and more efficiently than that of NXS sites [8]. Cherepanova et al. reported that the oxidoreductase activity of MAGT1 on OST-B is necessary for efficient N-linked glycosylation at the acceptor site with a proximal Cys residue, especially sites with Cys in the acceptor glycosite (NCT/S) [7]. In addition, Cherepanova et al. used glycoproteomics to quantitatively label approximately one thousand deglycopeptides [9]. The results showed that N-linked glycosylation on the short loop of multiple transmembrane proteins depends on the OST-B complex, and sites located in Cys-rich protein domains are more likely to be glycosylated by OST-B. Moreover, we realized that the glycotransferases and glycosidases not only changed the protein glycosylation but also regulated the proteins expression. Most importantly, the site-specific glycosylation, including the glycan structure and composition of specific glycosites, were also changed in the mutant cells [10,11]. Site-specific intact glycopeptide (IGP) analysis is widely considered to be the most promising strategy to characterize glycoproteins, which could provide a comprehensive snapshot of the glycoproteins, their glycosites, and the glycans occupying these sites at the same time [12].

Defects in STT3A or STT3B genes cause congenital disorders of glycosylation (CDG) [13,14]. However, it is not clear which proteins are primarily affected by the defects in STT3A or STT3B, and little is known regarding the relationship between STT3A and STT3B. In this study, proteome and site-specific glycoproteome analyses were used to explore proteins, pathways, and patterns of glycosylation affected by the deletion of STT3A or STT3B genes. In total, over three thousand global proteins and approximately two thousand intact glycopeptides (IGPs) were identified in STT3A-KO, STT3B-KO, and wild-type (WT) HEK293 cells. We demonstrated that the deletion of STT3A or STT3B exerts various effects on specific protein expression, heterogeneity of glycosylation, glycosites, and modification of N-glycans on IGPs.

## 2. Materials and Methods

### 2.1. Cell Culture and Protein Extraction

In our previous study, we generated STT3A-knockout (KO) and STT3B-KO cell lines in human embryonic kidney 293 (HEK293) cells using clustered regularly interspaced short palindromic repeat (CRISPR)-Cas9 gene editing technology [10,15,16]. KO and WT cells were cultured in DMEM (Biological Industries) containing 10% fetal bovine serum at 37 °C and 5% CO_2_. When the cells reached 80% confluence in a 100 mm dish, proteins were extracted as previously described [17]. In short, 6 × 10^6^ cells were washed with 1 × phosphate-buffered saline (PBS) for 2 min, followed by 8 M urea and 1 M ammonium bicarbonate buffer on the dish and lysed for 30 min on ice. Subsequently, the cell lysates were sonicated (SCIENTZ, JY92-IID) for 5 min, and centrifuged at 15,000× *g* for 20 min at 4 °C. The supernatant was collected and stored at −80 °C. Ten microliters of the samples were used to determine the protein concentration using a BCA assay kit (Beyotime Biotechnology, China). In this study, five biological replicates were separately performed in the cell culture and protein extraction, and the proteins were digested and (glyco)peptides were enriched separately for the mass spectrometry analysis, as described below.:

### 2.2. Protein Digestion and Intact N-Linked Glycopeptides Enrichment

Proteins (500 μg) were reduced using 10 mM DTT at 56 °C for 45 min and alkylated with 20 mM iodoacetamide in the dark for 30 min at room temperature. The solutions were diluted 8-fold in 40 mM ammonium bicarbonate, and sequencing grade trypsin (Beijing Shengxia, China) was added to the solution at 1:40 (*w*/*w*), followed by incubation overnight at 37 °C. The peptides were desalted using a C18 Sep-Pak column. Twenty micrograms of desalted peptides were reserved for peptide detection, and the remaining peptides were subjected to the enrichment of IGPs using the Oasis MAX extraction cartridges (Waters), as previously described [18].

### 2.3. LC-MS/MS Analysis

Lyophilized global peptides and glycopeptides were resuspended in 2% ACN/0.1% formic acid (FA) solution and analyzed using a high-resolution Orbitrap Fusion Lumos mass spectrometer (Thermo Scientific, San Jose, CA, USA). For each run, 1.5 μg peptides or 2 μg glycopeptides were applied to MS analysis. Five replicate experiments were conducted for proteomics and glycoproteomics analyses individually. Samples were separated on an EASY-nLC 1200 system (Thermo Scientific, San Jose, CA, USA) equipped with an internal RSLC C18 column (75 μm × 25 cm). The mobile phase consisted of phase A (0.1% FA) and phase B (90% ACN/0.1% FA). The flow rate was maintained at 550 nL/min. For global peptides, the elution gradient was set as follows: 0–8 min, 3% B; 8–88 min, 8% B; 88–110 min, 28% B; 110–115 min, 32% B; 115–120 min, 80% B. Other mass spectrometry parameters were performed according to methods previously established in our laboratory [19]. For glycopeptides, the gradient elution was 2–6% B, 1 min; 6–30% B, 90 min; 30–38% B, 22 min; 38–80% B, 5 min; and 80% B, 5 min. Spectra were collected from 350 to 2000 *m*/*z* at a resolution of 60,000 using an AGC target of 4 × 10^5^ and a maximum injection time of 250 ms. Secondary mass spectrometry analysis was then performed using the data-dependent HCD fragmentation pattern. The parameters were as follows: resolution of 60,000, collision energy of 35% for HCD, 5% for stepped collision, AGC target 5 × 10^4^, maximum injection time 150 ms, and microscan 1.

### 2.4. Data Analysis

For global peptides, the acquired MS/MS spectra were matched using MaxQuant software. The database uses HUMAN-UniProt-organism containing 20,432 proteins, updated on 22 July 2019. The search criteria were a fragment ion mass tolerance of 0.02 Da and a precursor ion tolerance of 10 ppm. Iodoacetamide derivatives of cysteine were set as fixed modifications, and deamidation of asparagine and glutamine and oxidation of methionine were set as variable modifications. The false discovery rate (FDR) for both PSM and protein was 0.01. For data normalization, the relative abundance of proteins was the ratio of valued individual protein intensity to the sum of total protein intensity in each replicate. Then, the mean of relative abundance of each protein identified in at least three replicates was taken for quantitative analysis.

For IGPs identification, the data were searched using GPQuest 2.0 [20] and a database containing 45,491 glycopeptides and 252 N-glycans [21], respectively. The mass tolerance parameters for MS^1^ and MS^2^ are 10 and 20 ppm, respectively. In addition, the matched results were filtered based on the following conditions: (1) FDR less than 1% for glycopeptides; (2) at least one N-glycan for one peptide spectra match (PSM). For the glycoproteomics, the glycopeptides identified in at least three replicates were selected for subsequent quantitative analysis. The normalization of relative abundance was based on the IGPs and the method was consistent with peptides. The relative abundances of glycoproteins, glycopeptides and IGPs in each sample were the mean of its relative abundance of parallel replicates, respectively.

DAVID bioinformatics resources were used to annotate the function of differentially expressed proteins (DEPs) and glycoproteins (DEGPs) in KO cells, including GO (BP: biological process, CC: cellular component, MF: molecular function) and KEGG analysis [22,23]. The enriched data filter criteria were performed as follows: FDR ≤ 0.05, *p*-Value ≤ 0.05. The pLogo was used for the prediction of glycosites [24]. The STRING was used to analyze protein interactions [25]. The quantitative results of Western blotting were obtained by ImageJ (2.1.0/1.53c, NIH, Bethesda, MD USA) software. Finally, the figures were prepared with R (4.1.1) (ggplot2, pheatmap package), Adobe Illustrator (2019), and Excel (2019) software. All the significance changes in histograms are displayed as the mean ± SD from independent experiments with p values (one-way ANOVA followed by Dunnett’s multiple comparisons test). ****, *p*-value < 0.0001; ***, *p*-value < 0.001; **, *p*-value < 0.01; *, *p*-value < 0.05.

### 2.5. Immunoblotting

Cells were cultured in 6-well dishes, harvested with EDTA-containing trypsin, centrifuged at 800× *g* for 3 min and washed twice with PBS. After discarding the supernatant, the cells were lysed in 40 μL of 1% NP40 containing 50 mM Tris-HCl, pH 7.5, 150 mM NaCl, 1% Triton X-100, 1 mM EDTA, and protease inhibitor cocktail (EDTA-Free, MCE) for 30 min on ice. Subsequently, the insoluble fraction was removed by centrifugation at 15,000× *g* for 10 min at 4 °C, and the supernatant was transferred to a new tube with SDS sample buffer and boiled at 95 °C for 5 min. Samples were run on SDS-PAGE, followed by Western blotting analysis.

Rabbit monoclonal anti-BIP (C50B12, CST), rabbit monoclonal anti-PERK (D11A8, CST), rabbit monoclonal anti-IRE1α (14C10, CST), mouse monoclonal anti-ATF6 (66563-1-Ig, Proteintech), rabbit polyclonal anti-GRP94 (14700-1-AP, Proteintech), and mouse monoclonal anti-β-Tubulin (HC101-01, Transgene) were used as primary antibodies; goat anti-rabbit IgG, HRP (HS101, Transgene) and goat anti-mouse IgG, HRP (HS201, Transgene) were used as secondary antibodies. Core-fucosylated N-glycans were detected using Lens culinaris agglutinin (Bio-LCA, J-Chemical), followed by streptavidin-HRP (eBioScience). Thapsigargin (Tg) (100 nM; Sigma-Aldrich) was used for drug treatments.

## 3. Results

### 3.1. Protein Expression and Functional Classification of STT3A-KO and STT3B-KO Cells

To understand the impact of OST on cellular proteins in mammalian cells, we performed proteomic and glycoproteomic analyses using STT3A-KO and STT3B-KO HEK293 cell lines. Peptides prepared from WT, STT3A-KO, and STT3B-KO cells were desalted using a C18 column and analyzed by LC-MS/MS individually. A total of 3335, 3119, and 3622 proteins were identified from WT, STT3A-KO, and STT3B-KO cells, respectively (Appendix A). Over 75% of the proteins were overlapped among the five biological replications and the Pearson’s correlation of the five biological replications were all over than 0.80 (Figure 1a). Compared to the identified proteins in WT cells, 427 and 269 proteins were differentially expressed (fold change (FC) > 2 and *p*-Value < 0.05) in protein relative abundance in STT3A-KO and STT3B-KO cells, respectively. In addition, a hydrophilic MAX column was applied to enrich glycopeptides from each sample. The enriched glycopeptides were analyzed by LC-MS/MS and 283, 260, and 251 glycoproteins were identified in WT, STT3A-KO, and STT3B-KO cells, respectively (Appendix A). Over 60% of the glycoproteins were overlapped among the five biological replications and the Pearson’s correlation of the five biological replications were all over than 0.70 (Figure 1b). Among these glycoproteins, 47 and 29 glycoproteins were differentially expressed in these two KO cell lines. From the distribution analysis of protein FC, 13.67% of proteins and 18.39% of glycoproteins in STT3A-KO cells exhibited downward trends (Figure 1c), and the total number of differentially expressed proteins (DEPs) and differentially expressed glycoproteins (DEGPs) identified in STT3A-KO cells was greater than those of STT3B-KO cells, indicating that the deletion of STT3A has a greater impact on intracellular protein expression than that of STT3B removal.

Annotation analysis showed that deletion of STT3A or STT3B remarkedly affected cellular pathways and molecular functions in the cells (Appendix A). KEGG pathway analysis indicated that “Protein processing in endoplasmic reticulum”, “Nucleocytoplasmic transport”, and “Ribosome biogenesis in eukaryotes” were the three most affected pathways in STT3A-KO cells, and their enriched DEPs/DEGPs accounted for more than 11%, whereas in STT3B-KO cells, the most affected pathways were “Biosynthesis of amino acids” (12.00%), “Cardiac muscle contraction” (10.34%), and “Lysosome” (7.58%) (Figure 1d and Appendix A). However, in general, the proportion of DEPs/DEGPs in these enriched pathways was not high. In addition, the “Oxidative phosphorylation” pathway was enriched in both knockout cells (9.70% in STT3A-KO and 7.46% in STT3B-KO). In GO analysis, protein binding and RNA binding represented the two most affected molecular functions in STT3A-KO and STT3B-KO cells, and these DEPs and DEGPs were also broadly distributed in cells (nucleus, cytosol, membrane, and extracellular exosome) (Figure 1e).

### 3.2. Glycosylation Heterogeneity in STT3A-KO and STT3B-KO Cells

To analyze the effects of STT3A or STT3B on glycosylation preferences and glycan structures, we characterized the site-specific glycosylation of proteins by comprehensive analysis of intact glycopeptides (IGPs). In total, 2570 IGPs containing 190 N-glycans from 498 glycosites were identified in WT cells. A total of 2458 IGPs containing 176 N-glycans from 470 glycosites were identified in STT3A-KO cells. Moreover, 2291 IGPs containing 191 N-glycans derived from 427 glycosites were identified in STT3B-KO cells (Figure 2a and Appendix A). The modification of glycans at different sites on a single protein and the different glycans modified at a single glycosite (micro-heterogeneity) were the primary reasons for the complexity of glycoproteins. Due to these heterogeneities in glycosylation, it is challenging to precisely characterize glycoproteins. With respect to the micro-heterogeneity, 26.31% of glycosites in WT cells were modified with one to two different N-glycans, and 43.57% of glycosites had three to five glycans among the glycosite-containing peptides, similar with STT3B-KO cells (24.82% and 44.96%). In the STT3A-KO cells, the percentage of glycosites modified with one to two different N-glycans was increased (27.45%), whereas the percentage of peptides modified with three to five glycans was reduced (40.43%), suggesting that the deletion of STT3A could reduce the micro-heterogeneity of glycosylation (Figure 2b). In addition, the percentage of proteins with only one glycosite was increased in STT3B-KO cells and the proportion of proteins with two to four glycosites was reduced (Figure 2c). The percentage of proteins with four or more glycosites and glycosites with more than five glycan structures were unexpectedly increased in STT3A-KO cells. For example, in addition to commonly detected glycosites, two extra glycosites (N_132_ST and N_149_GS) were modified with N-glycans in extracellular sulfatase Sulf-2 (SULF2) in STT3A-KO cells. Furthermore, for the N_171_YT glycosite identified on SULF2, nine N-glycans were identified in STT3A-KO cells, while three N-glycans were detected at this site in both WT and STT3B-KO cells (Figure 2d). However, none of these changes were significant, which suggests some function overlapping or complementary mechanism between OST-A and OST-B. It has been reported that OST-B modifies OST-A-dependent glycosites in STT3A-KO cells and may also bind to translocons to mediate co-glycosylation to increase the efficiency of N-glycosylation [26]. We analyzed the relative abundance of STT3A and STT3B proteins in WT and KO cells. Interestingly, the relative abundance of STT3B protein was increased in STT3A-KO cells (Figure 2e), suggesting that STT3B compensates the STT3A function in the absence of STT3A protein.

### 3.3. Characterization of STT3A- and STT3B-Dependent Glycoproteins and Glycosites

In this study, 223 glycoproteins were commonly detected in both WT and STT3A-KO cells, and 215 glycoproteins were commonly observed in both WT and STT3B-KO cells. The relative abundance of cell surface glycoproteins such as CD63, CD166, and CD276 was increased in both STT3A-KO and STT3B-KO cells. On the other hand, several procollagen modification enzymes (GT251, PLOD1, PLOD2) and ECM-receptor interaction associated glycoproteins (LAMB1, ITA5, ITB1) were decreased in STT3A-KO cells (Appendix A). For the glycopeptides, 26 upregulated and 46 downregulated in STT3A-KO cells, and 17 upregulated and 20 downregulated in STT3B-KO cells, were identified. We next analyzed the changes of glycosites in DEGPs in knockout cells. The glycosites from the glycoproteins that changed in Appendix A were also detected in the analysis. For example, N_381_TS and N_96_TS from a glycoprotein GT251 were downregulated in STT3A-KO cells, N_268_CS from FUCO, N_116_AS and N_646_GS from PTK7 were downregulated in STT3B-KO cells, and the expression of N_130_HT from CD63 was increased in both knockout cells. In addition, we also noticed that LRP1 protein, which is involved in receptor-mediated endocytosis and phagocytosis, showed a significant increase in different glycosites in both knockout cells. In addition, N_197_IT from PLOD1 was downregulated in STT3A-KO cells, but up-regulated in STT3B-KO cells, which are potential STT3A- or STT3B-dependent modification sites (Figure 3a and Appendix A).

To determine the sequence specificity of STT3A and STT3B recognition, the motif and positions of these downregulated glycopeptides on the proteins were characterized in detail (Appendix A). First, the distance from the N-terminus or C-terminus to glycosites of each protein was analyzed, revealing that 10.87% of glycosites were located within 80 amino acids from the N-terminus and 6.52% glycosites were located within 80 amino acids from the C-terminus in the STT3A-KO cells. In contrast, 5.00% of glycosites located within 80 amino acids from the N-terminus and 10.00% glycosites located within 80 amino acids from the C-terminus were identified in STT3B-KO cells (Figure 3b). The reduction of more sites located within 80 amino acids from the N-terminus in the STT3A-KO cells suggested that those glycosites are primarily dependent upon STT3A. On the other hand, the reduction of those from the C-terminus in the STT3B-KO suggested that those glycosites are primarily dependent upon STT3B, consistent with previous reports [8]. To characterize the probability of site modification, the ten amino acids before and after the Asp of the glycosite were analyzed. According to the representative value of each amino acid residue, the His located at the N-2 site and the Asp located at the N-5 sites were enriched in the STT3A-KO-dependent glycopeptides, suggesting that OST-A might prefer peptide sequences, having those amino acid residues around the Asp of glycosites. Furthermore, it is noted that Cys located at the N + 1, N + 6, N−10 and N+9 glycosites were enriched in the STT3B-KO-dependent glycopeptides, which was consistent with previous conclusions [7] (Figure 3c).

In addition, we focused on those glycopeptides with reduced N-linked glycosylation in KO cells. By comparing proteome and glycoproteome data, we screened 15 glycopeptides from 12 glycoproteins in STT3A-KO cells and 13 glycopeptides from 10 glycoproteins in STT3B-KO cells, where the protein level was similar with WT cells, but the glycopeptide level decreased in KO cells (Figure 3d). The majority of these glycoproteins were associated with lysosome in STT3A-KO cells, while STT3B was preferred with ER-related glycoproteins (Figure 3e).

### 3.4. Alteration of the N-Glycan Profile and Glycosylation-Related Proteins in STT3A-KO and STT3B-KO Cells

OST-A guides the transfer of N-glycans to nascent polypeptides during protein translation, while OST-B is responsible for the supplementary glycosylation modification of proteins after translocation in the ER. Until now, no studies have shown that OST has any impact on subsequent glycan processing, but we noticed that the relative abundance of several glycotransferases was significantly changed in KO cells based on proteomics and glycoproteomics results, which may cause a change in the glycan patterns on glycoproteins. Therefore, we analyzed whether the deletion of STT3A or STT3B altered the modification of N-glycan structures on IGPs. We found that the percentages of mannosylated N-glycans and sialylated N-glycans were decreased, while the ratio of fucosylated N-glycans was unexpectedly increased in STT3A-KO cells. On the other hand, the glycan profiles of STT3B-KO cells were similar to those of WT cells (Figure 4a). Most fucosylated N-glycans that were upregulated in STT3A-KO cells have only one fucose residue in the structures, likely increasing the amount of core fucosylation (Figure 4b). Lens culinaris agglutinin (LCA) was used to detect core fucosylation of glycoproteins, which showed a slightly increase in LCA binding signal in STT3A-KO cells compared with WT cells (Figure 4c). On the other hand, GlcMan9GlcNAc2 structures were increased, while Glc3Man9GlcNAc2 and Man (6–9) GlcNAc2 structures were decreased in mannosylated N-glycan structures (Figure 4d). Our results suggest that the initial transferring of oligosaccharides to proteins affects subsequent processing of the N-glycans. Since the STT3A complex is bound to the translocon and most of the STT3B complex is not bound, this difference might change the stability of some glycan processing enzymes, folding sites of glycosylated proteins in the ER, and the ER exit sites in vesicular transport.

To comprehensively understand the reason for the N-glycan structure changes in STT3A-KO cells, we assessed the relative abundance of all proteins involved in N-linked glycosylation in the ER and Golgi that were identified in our proteomics and glycoproteomics analyses (Figure 4e). The heatmap showed that the relative abundance of some OST subunits (RPN2, RPN1, and DDOST) was decreased in STT3A-KO cells. In contrast, the expression of molecular chaperones localized in the ER (CALR, CANX, UGGT1, and MLEC) was increased. It was also observed that the expression of proteins related to protein folding (BIP, PDIA4, PDIA3, HYOU1, etc.) was increased (Figure 4e, upper panel), consistent with the KEGG pathway analysis results (proteins belonging to the “Protein processing in the ER” category was enriched in STT3A-KO cells) (Figure 1d). In particular, UGGT1 transfers a Glc residue to the Man9GlcNAc2 structures to generate GlcMan9GlcNAc2, which is recognized by CALR and CANX [27]. Upregulation of UGGT1 in STT3A-KO cells may cause an increase in GlcMan9GlcNAc2 structures. Moreover, transferring fewer N-glycans to nascent peptides due to defects in STT3A may affect the protein stability and function of some glycosyltransferases, which alters glycan profiles. For example, ER glucosidase I (MOGS) mediates the trimming of the terminal Glc from the Glc3Man9GlcNAc2 structure, which is the first step after the glycan is transferred to nascent polypeptides via OST. The significant downregulation of MOGS in STT3A-KO cells supports our hypothesis.

### 3.5. Hyperglycosylation in the STT3A-KO and STT3B-KO Cells

To our surprise, some neo-glycosites appeared only in STT3A-KO or STT3B-KO cells (Figure 5a and Appendix A). Compared to WT cells, 85 and 63 glycopeptides from 67 and 54 glycoproteins were only identified in STT3A-KO and STT3B-KO cells, respectively. More than 80% had only one additional glycosite on the proteins. For hyperglycosylation, we focused on glycoproteins with more than two unique glycosites (15 proteins) only detected in STT3A-KO or STT3B-KO cells, but not in WT cells (Figure 5b). First, a functional annotation and protein–protein interaction (PPI) analysis of these hyperglycosylated proteins was performed. These proteins, which mainly located on the membrane and secreted out of cells through vesicles, interacted closely according to PPI strength and were associated with ECM-receptor interaction, focal adhesion and PI3K-Akt signaling pathways. Then, we calculated the expression of these proteins in global and glycoprotein analyses (Figure 5c). Some proteins, such as NRCAM, SEL1L and LRP1, have an increased occupation of putative glycosite in both STT3A-KO and STT3B-KO cells. Conversely, ATRN, ECE1, ITGA1, and SULF2 were hyperglycosylated only in STT3A-KO cells, while the relative abundance of these proteins was increased in STT3A-KO cells and decreased in STT3B-KO cells, suggesting that these sites are hyperglycosylated by STT3B.

ENPL (Endoplasmin/HSP90B1/GRP94) is a molecular chaperone that plays a role in protein folding in the ER [28]. In a previous study, five silent N-glycosites of ENPL were reportedly abnormally glycosylated in STT3A-KO cells [9]. Hence, we also analyzed the N-linked glycosylation of the protein, and two additional glycosites (N_107_AS and N_445_VS) were detected only in STT3A-KO cells (Figure 6a), indicating that OST-B transfers N-glycans to those sites in defect of OST-A. In addition to the altered numbers of glycosites, the profiles and abundance of N-glycans at different sites in KO cells were also changed. In STT3A-KO cells, the relative abundance of N-glycans at glycosite N_217_DT was significantly reduced. Additional glycosylation sites in STT3A-KO are modified not only with mannosylated N-glycans but also with fucosylated or sialofucosylated complex-type N-glycan structures (Figure 6a). Nevertheless, the relative abundance of ENPL in STT3A-KO cells was downregulated both in the global MS (Figure 5c) and Western blotting (Figure 6b upper) result. From the Western blotting, we also can see that the migration of the ENPL was slowed in STT3A-KO cells. As mentioned above, molecular chaperones (UGGT1, CALR, CANX et al.) were upregulated in STT3A-KO cells, which could be caused by ER stress. Thapsigargin (Tg), an inhibitor for sarco/endoplasmic calcium ion ATPase activity, induces ER stress in cells. When we treated WT cells with Tg for 24 h, a fraction of ENPL proteins was also shifted, similar to the deletion of STT3A. Subsequently, we treated the protein samples with Endo H to excise the N-glycans on the protein, and the molecular weight of ENPL was decreased, demonstrating that the molecular weight shift was caused by the modification of N-glycans (Figure 6b bottom). The abundance decreased but the glycosylation increased of ENPL in STT3A-KO cells, which was consistent with a previous report that ENPL is highly glycosylated under ER stress conditions [29]. This can be explained by hyperglycosylated ENPL forming an unnatural conformation and exhibiting reduced activity, which is degraded through OS-9-mediated, but ERAD-independent, lysosomal degradation pathways. Therefore, we speculate that some stress pathways were activated with the deletion of STT3A. In fact, expressions of BIP, ATF6 and PERK were increased in STT3A-KO cells, as shown in Western blotting, similar to WT cells treated with Tg (Figure 6c). These results indicated that the ER stress and unfolded protein response (UPR) occurs when the STT3A is deleted from the cells. Therefore, we believe that the deletion of STT3A causes ER stress due to the impaired N-linked glycosylation, leading to the hyperglycosylation of related proteins, which is mediated by OST-B and degradation through the lysosome pathway.

## 4. Conclusions

In this study, we performed a comprehensive characterization of wild-type (WT) and STT3A and STT3B knockout (STT3A-KO and STT3B-KO) HEK cells using proteomics and glycoproteomics. It was found that STT3A and STT3B affect biological functions in cells, especially for binding activities and metabolic pathways. Moreover, STT3A was the predominant oligosaccharyltransferase compared with STT3B in the mammalian cells. In addition, STT3A and STT3B show different sequence specificity on N-glycosites. The glycosite at the C-terminus and Cys-enriched glycosites are dependent upon STT3B, while STT3A is very prone to His residues at N—2 position, or Asp residue at N—5 position. However, even though the different glycosite preferences exist for STT3A and STT3B, they can compensate for each other and complement the glycosylation process in the absence of one of them. Interestingly, we found remarkable hyperglycosylation regarding the relative abundance of glycans and glycosites in KO cells, especially in STT3A-KO cells. Furthermore, the suppression of STT3A causes ER stress and activates the UPR pathway. Currently, we speculate that the loss of OST could affect the glycosylation of these glycosylation-related enzymes which may hinder their functions, as well as impact some transcription factors that mediate endogenous protein expression.

In this study, we aimed to analyze general effects of STT3A and STT3B. We performed experiments using HEK293 cells as model cells because of their ease of handling and gene knockout. However, the cell lines are not suitable for the analysis of symptoms in CDG defective in OST complex. Additionally, most CDG patients are caused by point mutations in genes. The symptoms vary depending on the location of the mutation and the substituted amino acid. The KO cells we used are assumed to be the most severe case. It would be useful if specific cells from tissues affected in STT3A-CDG and STT3B-CDG patients are used to clarify the relationship with the disease. In the future analysis, we expect to apply the methods to specific cells, such as hepatocytes, based on the results of this study.

In all, our work revealed that OST-A and OST-B not only regulated glycosite-dependent N-linked glycosylation differentially, but also varied protein expression and N-linked glycan modifications intracellularly.

## Figures and Tables

**Figure 1 cells-11-02775-f001:**
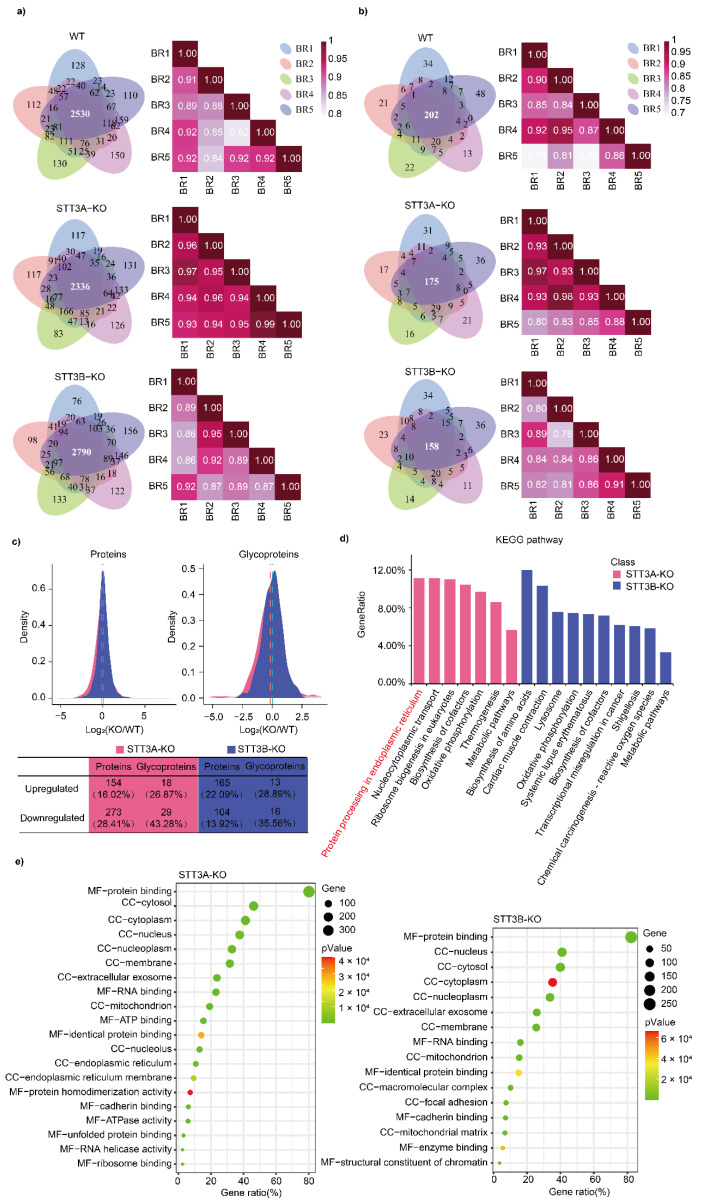
Identification and functional classification of proteins in STT3A-KO and STT3B-KO cells compared to wild-type (WT) HEK293 cells. Overlapping and Pearson’s correlation relationship of proteins (**a**) and glycoproteins (**b**) in five biological replicates (BR). (**c**) Density distribution plot of the relative abundance of identified proteins and glycoproteins in KO cells compared to WT cells. Pink, STT3A-KO cells; blue, STT3B-KO cells. Percentages of proteins and glycoproteins with two-fold changes between KO and WT cells (fold change (log2) larger than 1 and less than −1, *p*-Value < 0.05) are listed. (**d**) Kyoto Encyclopedia of Genes and Genomes (KEGG) enrichment of differentially expressed proteins (DEPs). The *Y*-axis represents the number of enriched DEPs in each pathway. (**e**) Bubble plots depicting the top ten Gene Ontology (GO) annotations for DEPs in WT and KO cells. The size and color of the bubbles reflect the number of genes enriched and the *p*-Value, respectively. The *X*-axis coordinates indicate MF: molecular function; CC: cellular component.

**Figure 2 cells-11-02775-f002:**
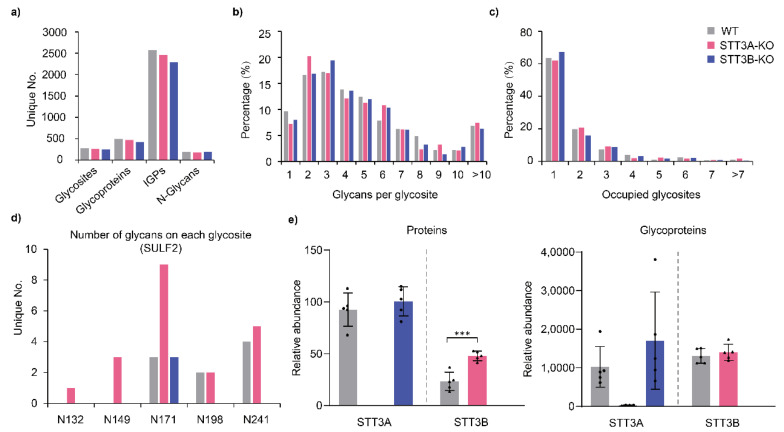
Glycoprotein analysis in WT and KO cells. (**a**) Numbers of identified glycosites, glycoproteins, IGPs, and N-glycans. Numbers of glycans per glycosite (**b**) and glycosites per protein (**c**). (**d**) Number of glycans on each glycosite of extracellular sulfatase Sulf-2 (SULF2). (**e**) Relative abundance of STT3A and STT3B proteins in WT and KO cells. STT3A and STT3B protein abundance in global protein (left) and glycoprotein (right) analyses are shown as bar plots. ***, *p*-value < 0.001.

**Figure 3 cells-11-02775-f003:**
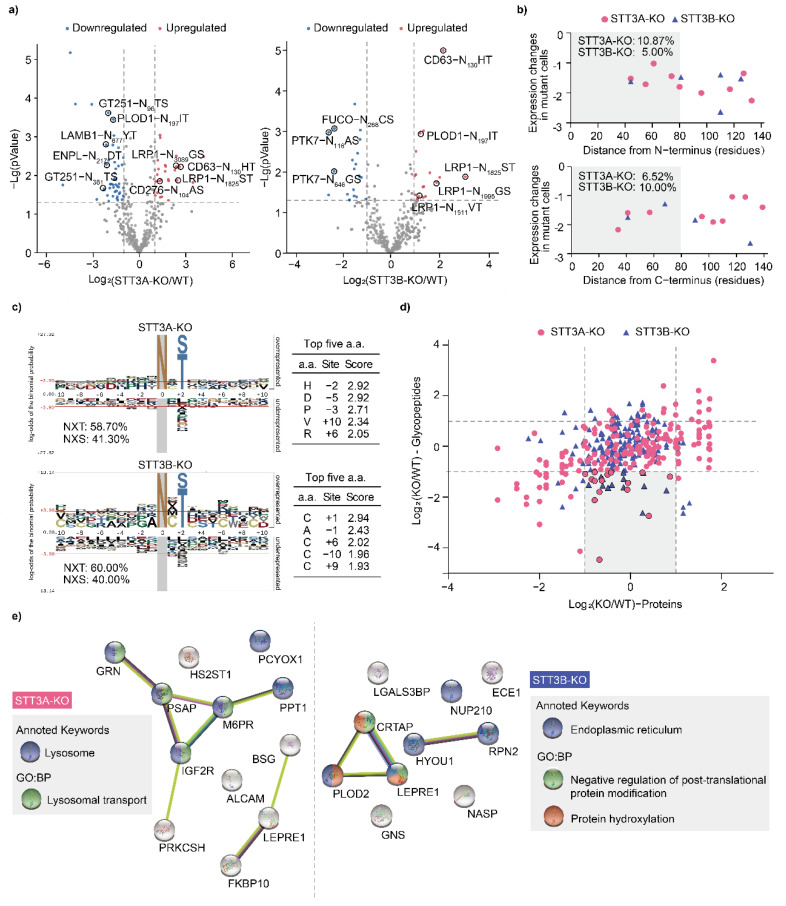
Differential expression of glycopeptides and N-glycosylation preference between WT and KO HEK293 cells. (**a**) Volcano plot of glycopeptides expression. The dashed line represents a two-fold change (*X*-axis) and *p*-Value < 0.05 (*Y*-axis) in KO cells compared to WT cells. (**b**) Scatter diagram of N-glycosites. The distance of the N-glycosites from the N-terminus (upper) or C-terminus (bottom) of the proteins is shown. The *Y*-axis values represent changes in glycopeptide abundance (log_2_) between KO cells and WT cells. The percentages of affected N-glycosylation sites within 80 amino acids from the N-terminus or C-terminus are listed. (**c**) Frequencies of ten amino acids before and after the N-glycosite in glycopeptides. The top five amino acids around N-glycosylation sites with high probability in STT3A-KO and STT3B-KO cells are listed. (**d**) Scatter diagram of N-glycosites and correlated proteins expression in KO cells compared with WT cells. (**e**) Protein–protein interaction (PPI) and annotation of glycoproteins with decrease N-linked glycosylation in STT3A-KO (left) and STT3B-KO (right) cells.

**Figure 4 cells-11-02775-f004:**
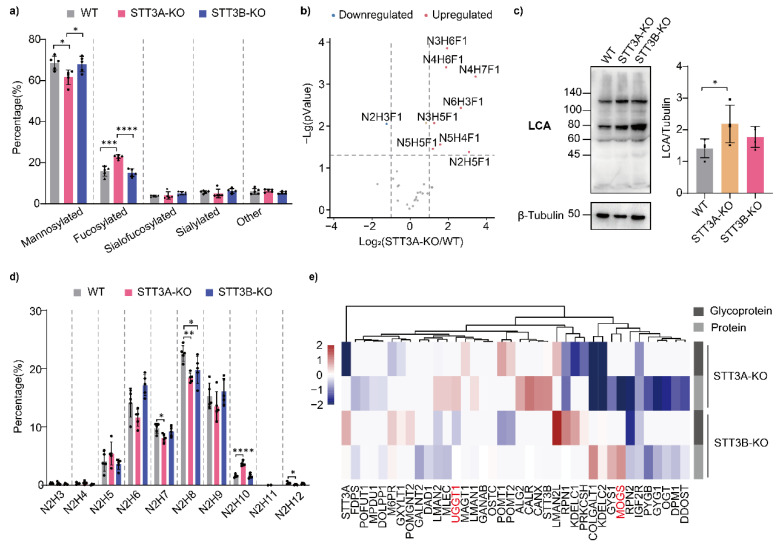
Relative abundance of N-glycan profiles and N-glycan-related proteins in STT3A-KO and STT3B-KO compared to WT HEK293 cells. (**a**) Percentage of different N-glycans. Mannosylated: paucimannose, oligomannose-type and Glc (1–3) Man9GlcNAc2 N-glycans; fucosylated: N-glycans modified by fucose, but not sialic acid; sialofucosylated: N-glycans modified by both fucose and sialic acid; sialylated: N-glycans modified by sialic acid, but not fucose; other: hybrid and complex type glycans with no fucose or sialic acid modification. (**b**) Volcano plot of fucosylated glycans in STT3A-KO cells. N: N-acetylhexosamine; H: hexose, F: fucose. (**c**) Lectin blotting using LCA and the quantitation in WT and KO cells. LCA: Lens culinaris agglutinin. ****, *p*-value < 0.0001; ***, *p*-value < 0.001; **, *p*-value < 0.01; *, *p*-value < 0.05. (**d**) Relative abundance of all identified mannosylated N-glycans. (**e**) Expression of N-glycan-related proteins detected in proteins and glycoproteins. The color indicates log_2_ values of relative abundance changes in KO cells compared to WT cells. Red indicates upregulated, while blue indicates downregulated. Scales were limited from −2 to 2.

**Figure 5 cells-11-02775-f005:**
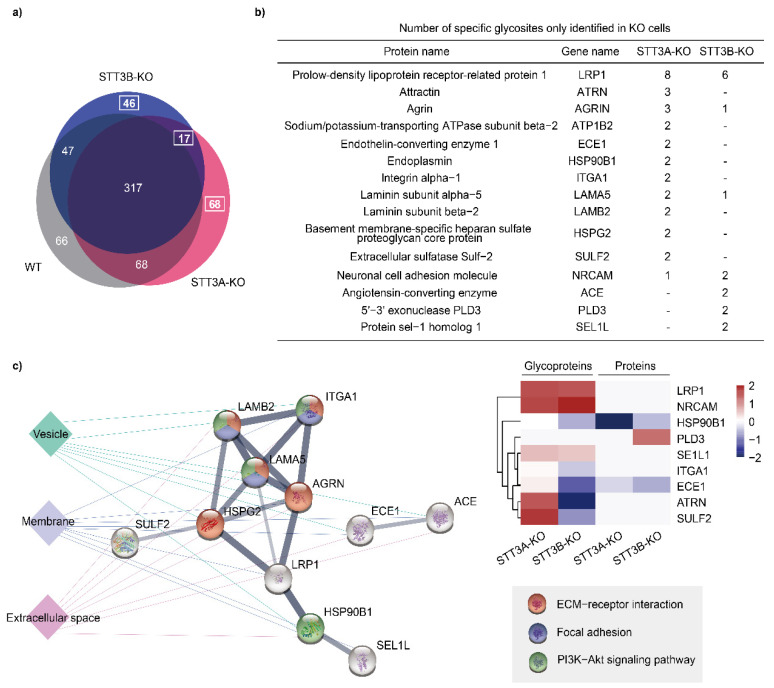
Hyperglycosylated proteins in STT3A-KO and STT3B-KO cells. (**a**) Venn diagrams of glycopeptides identified in WT and KO cells. (**b**) Proteins with at least two additional glycosites in KO cells are listed. Numbers represent the additional glycosites on the proteins observed in STT3A-KO and STT3B-KO cells. (**c**) The protein–protein interaction (PPI) network and cellular localization of hyperglycosylated proteins (left), and their relative abundance in proteins and glycoproteins in KO cells compared with WT cells (right). The thickness of the lines in the PPI represents the strength of the correlation, and the colors indicate different intracellular localizations and pathways.

**Figure 6 cells-11-02775-f006:**
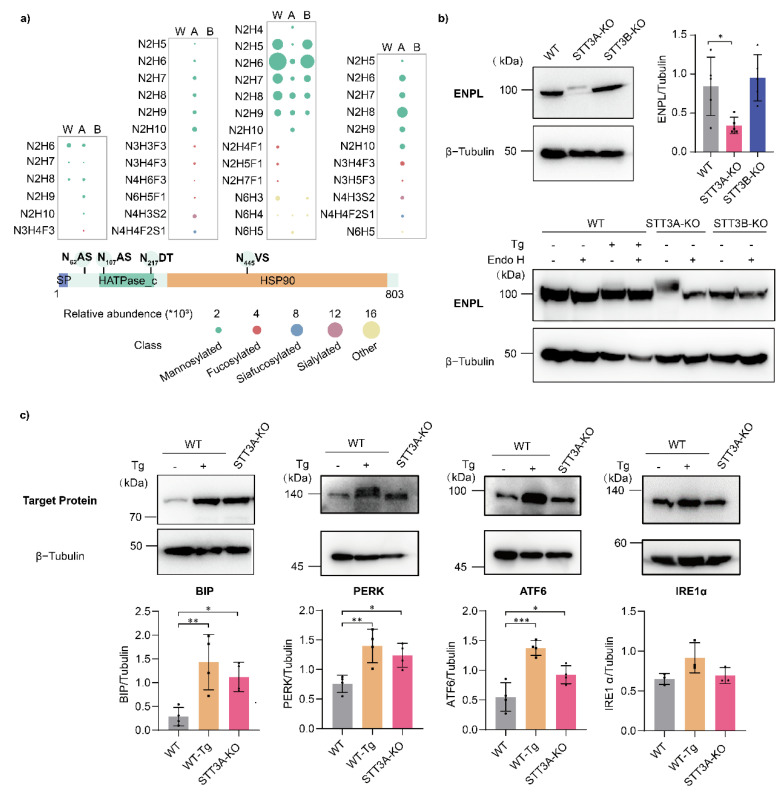
IGP analysis and biological verification in WT and KO cells. (**a**) IGP analysis of ENPL. The size of bubbles represents the relative abundance, and the color represents the type of glycan. (**b**) Western blotting of ENPL in WT and KO cells. Thapsigargin (Tg) treatment: 100 nM for 24 h. Endo H treatment for 1 h. W: WT; A: STT3A-KO; B: STT3B-KO. (**c**) Expression of BIP, PERK, IRE1α, and ATF6 in WT cells with or without thapsigargin (Tg) (100 nM for 24 h) and STT3A-KO cells was detected using immunoblotting. β-Tubulin was used as a loading control. ***, *p*-value < 0.001; **, *p*-value < 0.01; *, *p*-value < 0.05.

## Data Availability

All data generated or analyzed in the current study are included in this article. The mass spectrometry data have been deposited to the ProteomeXchange Consortium via the PRIDE partner repository with the dataset identifier PXD031276.

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
