# Peer review of "Proteome and Glycoproteome Analyses Reveal the Protein N-Linked Glycosylation Specificity of STT3A and STT3B"

_cells, 2022, doi:10.3390/cells11182775_

Round 1

Reviewer 1 Report

The article describes the combined use of proteomics and glycoproteomics in order to identify the effect of knockout of the two catalytic subunits of the oligosaccharyltransferase complex, STT3A and STT3B. It should be noted that in 2019, Cherapanova et al. performed similar work on the same cell model (HEK293 cells), but excluding the proteomics aspect. The authors acknowledge this prior work so this does not disqualify the findings of their submission, although it does somewhat reduce novelty.

The goal itself is a useful one, and much is unknown about the specificity of these subunits that appear to perform similar and partially overlapping roles. The authors could have done more with the data as presented. Conclusions are often rather superficial without much explanation of what these findings could mean in a cell biological fashion. However, there are also some important findings in STT3A/B-KO cells, such as the preference of STT3A for certain motifs, confirmation of hyperglycosylation of certain protein substrates, and the apparent increase in ER stress on STT3A-KO.

Sentences are often a little confusing and the exact meaning has to be inferred from previous sentences. This makes the text at times hard to follow. For example, in line 353 the authors state that ‘[proteins], were increased in both STT3A-KO and STT3B-KO cells due to their additional glycosites’. However, it seems that they really mean these proteins have an increased occupation (or glycosylation) of putative glycosites. This is just one example of many and the article would benefit from a thorough proofing before publication to improve clarity.

Comments:

i)                    Lines 181-183. What does metabolic pathway mean in this context? Which metabolic pathways are affected? Are all metabolic pathways affected? This would seem far too broad for any useful conclusion. A more detailed explanation of what this means should be included. Additionally, it could just be that overall, more proteins are included in this KEGG pathway. It would then be logical that you would find more DEPs there. Perhaps a better measure would be to divide the affected proteins by the total number of proteins within that subset of KEGG pathway proteins.

ii)                  Lines 223-224 & figure 2D: I Disagree with the statement that STT3A protein expression was upregulated in STT3B KO HEK cells, as stated. The difference is almost nothing, and certainly not significant, with a large overlap between the two. I would say that STT3A protein levels are in fact quite conclusively unaffected by STT3B knockout, given these results. This is not necessarily unexpected. Perhaps STT3A is only able to glycosylate cotranslationally and does not compensate for the missing STT3B in any meaningful way.

iii)                Lines 191-192: I don’t understand how, for example, 2,570 IGPs were detected but N-glycans were detected from only 498 glycosites. Surely there has to be at least one glycosite per IGP?

iv)                I am surprised that no mention was made of the downregulation of MAGT1 in STT3A-KO (see Fig4e). This is an unexpected finding and should be mentioned in the text – MAGT1 is thought to be a OST-B specific subunit. Why then, would we see It downregulated when less OST-A is present? This is particularly strange when taken alongside the data in fig2d, where it appears that STT3B is upregulated in STT3A-KO cells.

v)                  Some effort is put into studying the microheterogeneity at singular glycosites, meaning the presence of different N-glycans at a single site. However, at the point at which STT3A or STT3B is active, it is commonly accepted that only (or mostly) the N-glycan from the fully formed intact LLO is transferred onto glycoproteins. This N-glycan is then modified further downstream in the ER and Golgi. Differences then, in the ‘microheterogeneity’ at glycosites would indicate that a lack of N-glycan transfer by STT3A or STT3B leads also to differential modelling downstream. How would this work? There is some discussion of this (lines 295-296) stating that transfer of N-glycans to nascent peptides was not completely performed, but this seems self-evident and the single most expected effect of STT3A-KO, and to me an inadequate explanation for altered downstream N-glycan modelling. This is potentially important finding for the field of Congenital Disorders of Glycosylation, if it links the pathogenic mechanisms of CDG-I and CDG-II. Some further discussion is warranted.

vi)                Figure 6a is a nice representation of the data but it could be made clearer which box is referring to which glycan. I assume they follow the order of the N-glycans on the glycoprotein, but this is not obvious in the figure as displayed.

vii)               Figure 6b, it’s perhaps a bit simplistic to say that ENPL is simply downregulated. It could be that the hyperglycosylation is covering an epitope that the antibody usually recognises. Do you have data on the epitope to which the antibody was raised? Indeed, there is also a new band slightly above those in the WT and STT3B-KO cells, this could correspond to ENPL with additional glycan(s)? This possibility should at least be mentioned.

Minor comments:

i)                    Line 57, change congenital glycosylation disorders to Congenital Disorders of Glycosylation, to fit with the acronym CDG.

ii)                  Line 115, remove the .0 from 10.0

iii)                Lines 210-212: To make this clearer, please change ‘one and two N-glycans’ to ‘one to two different N-glycans’. This makes the reader more easily understand that you are discussing the heterogeneity of N-glycans at one glycosite.

iv)                Figure 2C: This should be changed to ‘occupied glycosites’ to make it more clear that the occupation of these glycosites is being measured.

v)                  Lines 238-239: Do the authors mean that 223 and 215 DEGPs were detected? Or overall glycoproteins. They first state glycoproteins then discuss DEGPS. This should be clarified.

vi)                Line 271: Remove the word glycosite here.

vii)               Figure 4b, ‘unrugulated’ should be unregulated

Reviewer 2 Report

The authors evaluated wild-type and STT3A and STT3B knockout HEK cells using proteomics and glycoproteomics. This is a thorough and very important study on one of the most important protein super complex in glycosylation.

 The most interesting finding was that they found STT3A, as the predominant oligosaccharyltransferase in their cell models. The two models (STT3A and STT3B KO) also showed different sequence specificity on N-glyco-sites but were able to compensate for each other’s missing function.

Surprising that besides significant hypoglycosylation they also found hyperglycosylation.

The major comment I have is on the disease model.

First, kidney is not an organ affected by either STT3A and STT3B, and one would look for a cell model with closer relevance for the human disease, like a liver cell based model (actually the most important for secretory N-glycosylation)

The second important criticism is the KO model. Most patients with either STT3A and STT3B defect are diagnosed with missense mutations, and only very few patients have loss of function variants.

How far this model is reliable to teach us about the human disease?

The proteomics data showed that the relative abundance of some OST subunits (RPN2, RPN1, and DDOST) was decreased in STT3A-KO cells. This is surprising, as one would suspect an upregulation of other subunits, especially for compensatory function. I think that the authors should check the gene expression of these few genes, also the STT3B gene, and the major upregulated chaperons, like UGGT1, to have a clear message and hypothesis.

The authors suggest that upregulation of UGGT1 in STT3A- 321 KO cells may cause an increase in GlcMan9GlcNAc2 structures. This is an important question, and the authors could measure this by LLO analysis or additional glycomics, especially as MOGS is downregulated.

What could be the reason, based on the metabolic differences between the STT3A and ATT3B knockouts that only STT3A suppression caused ER stress?

Would the authors conclude that the STT3B subunit is “rudimentary” in its function and most of the relevant glycosylation regulation tasks are related to STT3A?

Minor comment

Please clarify the sample harvesting from the cell models for glycoproteomics, the actual harvesting technique is only described under Western blot (EDTA rich tripsine), but I assume this is not the method the authors used for Proteomics/glycoproteomics?
